# Structure and Magnetic Properties of AO and LiFePO_4_/C Composites by Sol-Gel Combustion Method

**DOI:** 10.3390/molecules28041970

**Published:** 2023-02-19

**Authors:** Kaimin Su, Fang Yang, Qian Zhang, Huiren Xu, Yun He, Qing Lin

**Affiliations:** 1College of Biomedical Information and Engineering, Hainan Medical University, Haikou 571199, China; 2College of Physics and Technology, Guangxi Normal University, Guilin 541004, China

**Keywords:** LiFePO_4_, ZnO, MgO, composite, Li-ion battery, magnetic, cathode material

## Abstract

LiFePO_4_ takes advantage of structure stability, safety and environmental friendliness, and has been favored by the majority of scientific researchers. In order to further improve the properties of LiFePO_4_, AO-type metal oxides (MgO and ZnO) and LiFePO_4_/C composites were successfully prepared by a two-step sol-gel method. The effects of AO-type metal oxides (MgO and ZnO) on LiFePO_4_/C composites were studied. TG, XRD, FTIR, SEM and VSM analysis showed that the final product of the MgO and LiFePO_4_/C composite was about 70.5% of the total mass of the precursor; the complete main diffraction peak of LiFePO_4_ and MgO can be found without obvious impurity at the diffraction peak; there is good micro granularity and dispersion; the particle size is mainly 300 nm; the saturation magnetization (Ms), the residual magnetization (Mr) and the area of hysteresis loop are increased with the increase in MgO content; and the maximum Ms is 11.11 emu/g. The final product of ZnO and LiFePO_4_/C composites is about 69% of the total mass of precursors; the complete main diffraction peak of LiFePO_4_ and ZnO can be found without obvious impurity at the diffraction peak; there is good micro granularity and dispersion; the particle size is mainly 400 nm; and the coercivity (Hc) first slightly increases and then gradually decreases with the increase of zinc oxide.

## 1. Introduction

With the shortage of disposable energy, green and environment-friendly lithium-ion secondary batteries have become the focus of attention [1,2]. Lithium-ion batteries have been well used in various electrical equipment because of their advantages such as high energy density, small self-discharge current, many cycles and long service life [3]. Examples include emergency lights, flashlights, cameras, mobile phones, music players, video players, video game consoles, electronic computers, electric toys, hybrid electric vehicles, trams, robots and UAVs, etc. With the increasing demand for lithium-ion batteries, it is necessary to further improve the capacity and power of lithium-ion batteries. At present, lithium iron phosphate, as the cathode material of lithium-ion batteries, is one of the hotspots in this research field.

Since 1997, J.B. goodenough et al. [4] has proposed that lithium iron phosphate (LiFePO_4_) is a cathode material (cathode material) for lithium-ion batteries. As LiFePO_4_ has the advantages of having a long service life and rich raw materials; is non-toxic, pollution-free, and environment-friendly; and has good cycle performance, good safety performance, high capacity, high voltage, a low self-discharge rate and low price [5], it is loved and deeply studied by researchers. In the crystal structure of olivine lithium iron phosphate, all the atoms are arranged in a hexagonal close arrangement, but there is a slight distortion. It is an orthogonal crystal system. The crystal structure space group is Pnma [4], in which the phosphorus atom (P) and four oxygen atoms (O) form the PO_4_ structure, and the lithium atom (Li) and iron atom (Fe) form the octahedral structure of LiO_6_ and FeO_6_ with the surrounding six oxygen atoms (O), respectively. Below the Neel temperature, the magnetic properties of LiFePO_4_ are antiferromagnetic, which is caused by the super exchange between internal iron atoms (Fe) through the Fe-O-P-O-Fe bond [6].

At present, research on the influence of AO-type metal oxides on lithium iron phosphate (LiFePO_4_) is mainly about its electrical properties. For example, B. León et al. [7] optimized the electrochemical properties of lithium iron phosphate by introducing zinc oxide to prepare lithium iron phosphate composites. Shuxin Liu et al. [8] improved the electronic conductivity of lithium iron phosphate by modifying lithium iron phosphate with zinc oxide. Zhang et al. [9] studied a three-dimensional (3D) nanostructured skeleton substrate composed of hollow carbon fiber/carbon nanosheet/ZnO and found that the full cell of a NHCF/CN/ZnO/Li anode with LiFePO_4_ can work very well. Qiu X et al. [10] studied and found that full cells with LiFePO_4_ cathodes sustain 300 cycles with 98.8% capacity retention at 1 C by pairing with the Li-CuO@Cu anodes. ZnO and carbon co-modified LiFePO_4_ nanomaterials (LFP/C-ZnO) were prepared by Xiaohua Chen et al. [11], and it was found that when x = 3 wt%, LFP/C-xZnO exhibited well-dispersed spherical particles and remarkable cycling stability (it maintained 94.8% of the initial capacity after 50 cycles at 0.1 C). However, there are few studies on the magnetism of lithium iron phosphate (LiFePO_4_). Among them, the essence of the antiferromagnetism of lithium iron phosphate (LiFePO_4_) was studied by Jun Sugiyama et al. [12]. The hysteresis loop of LiFePO_4_/carbon at room temperature was studied by Ming Chen et al. [13] and it can be showed that the hysteresis loop of some lithium iron phosphate at room temperature had obviously weak ferromagnetism. The LiFePO_4_ (010) surface, due to exposure to atmospheric gases, was studied by Karol jarolimek and Chad RISKO [14], and it can be found that the molecular additives have an impact on the LFP magnetic properties. Wensong Zou et al. [15] prepared magnetic adsorbent from waste spent LiFePO_4_ batteries and found that the adsorption capabilities achieved for Cu^2+^, Cd^2+^, and Mn^2+^ were up to 71.23, 80.31 and 68.73 mg·g^−1^. Hongying Hou et al. [16] studied the standardization curves M (H) of the LiFePO_4_/C samples calibrated at 500 °C, 600 °C, 650 °C, and 700 °C.

From the above, it can be seen that the magnetic properties of lithium iron phosphate are closely related to the electrical properties. It is clear that the magnetic properties of lithium iron phosphate can further improve the electrochemical performance of lithium iron phosphate and provide a reference basis for further promoting the application of lithium iron phosphate in other fields. For example, using the magnetic detection of lithium iron phosphate and recycling the waste lithium iron phosphate. However, there is little research on the effect of AO-type metal oxides (MgO, ZnO) on the structure and magnetism of lithium iron phosphate (LiFePO_4_). Therefore, this paper mainly uses a two-step synthesis of AO-type metal oxides (MgO, ZnO) and LiFePO_4_/C composites, and studies the effect of different content of AO-type metal oxides (MgO, ZnO) on lithium iron phosphate carbon composites.

## 2. Results and Discussion

### 2.1. Thermogravimetric Analysis

In order to study the decomposition and quality changes of precursors of different contents of magnesium oxide and lithium iron phosphate carbon composites at high temperature, the precursors with a mass ratio of MgO to LiFePO_4_/C of 0.06 were analyzed by a thermogravimetric method, which was conducted in an N_2_ atmosphere. The thermogravimetric analysis (Figure 1) shows that there are three obvious mass change platforms at high temperatures for the composite precursor.

First, in the temperature range from room temperature to 110 °C, the heatflow curve shows an obvious endothermic peak, which mainly corresponds to the water evaporation endothermic reaction of the precursor; the DTG curve shows an obvious mass change peak; and the TG curve shows that the sample mass decreases by about 17.5%. Second, in the temperature range from 110 °C to 260 °C, the heatflow curve shows that there is a relatively small endothermic peak, which mainly corresponds to the detachment of crystalline water in the composite precursor [17]; the DTG curve shows that there is a relatively small mass change peak; and the TG curve shows that the sample mass is reduced by about 3%. Thirdly, from 260 °C to 400 °C, the heatflow curve shows that there is an obvious endothermic peak, which mainly corresponds to the decomposition of NO_3_^−^ and organic matter in the precursor [18]; the DTG curve shows that there is a relatively obvious mass change peak; and the TG curve shows that the sample mass is reduced by about 9%. In general, the decrease of the sample mass is mainly before 400 °C, which is similar to the TG curve of the magnesium-doped lithium iron phosphate prepared by Xu Heng Liu et al. [19]. After 400 °C, the mass of the precursor basically remains unchanged and tends to be stable. The final composite material is about 70.5% of the total mass of precursor, which is about 30% compared with the TG analysis product in reference [20] and about 40% in reference [21], with a higher relative yield. It shows that there may be target product synthesis at 400 °C, while the heatflow curve has an obvious endothermic peak at about 900 °C. The DTG and TG curves show that there is no mass change around this temperature, which may correspond to the growth of material crystal particles at high temperature [22], potentially resulting in large particles affecting material properties. Therefore, the more suitable temperature range for preparing materials may be about 400~900 °C.

Figure 2 is a thermogravimetric analysis of a precursor with a mass ratio of zinc oxide to LiFePO_4_/C composite of 0.10. Thermogravimetric analysis is performed in an N_2_ atmosphere. Figure 2 shows that there are three obvious mass change platforms of the composite precursor at high temperature. First, in the temperature range from room temperature to 100 °C, the heatflow curve shows an obvious endothermic peak, which mainly corresponds to the water evaporation endothermic of the precursor; the DTG curve shows an obvious mass change peak; and the TG curve shows that the sample mass decreases by about 16%. Second, in the temperature range of about 100 °C to 240 °C, the heatflow curve shows that there is a relatively small endothermic peak, which mainly corresponds to the detachment of crystalline water in the composite precursor [17]; the DTG curve shows that there is a relatively small mass change peak; and the TG curve shows that the sample mass is reduced by about 5%. Thirdly, from 240 °C to 410 °C, the heatflow curve shows that there is an obvious endothermic peak, which mainly corresponds to the decomposition of NO_3_^−^ and organic matter in the precursor [18].

The DTG curve shows that there is a relatively obvious mass change peak and the TG curve shows that the sample mass is reduced by about 10%. In general, the decrease of sample quality is mainly before 410 °C, which is similar to the TG curve of preparing a magnesium-doped lithium iron phosphate material. After 410 °C, the mass of the precursor basically remains unchanged and tends to be stable. The final composite material accounts for about 69% of the total mass of precursor, which is about 30% compared with the TG analysis product in reference [20] and about 40% in reference [21], with a higher relative yield. It shows that there may be target product synthesis at 410 °C, while the heatflow curve has an obvious endothermic peak around 900 °C. The DTG and TG curves show that there is no mass change around this temperature, potentially corresponding to the growth of material crystal particles at a high temperature, which may result in large particles affecting material properties [22]. Therefore, the more suitable temperature range for preparing materials may be about 410~900 °C.

### 2.2. XRD Analysis

Figure 3 is the XRD diagram of magnesium oxide and lithium iron phosphate carbon composites with different mass ratios. It shows that there are lithium iron phosphate peaks in each composite, and the main diffraction peaks, including (200), (101), (111), (211), and (311), correspond to the standard card (PDF#83-2092), indicating that each composite sample has a complete pure phase lithium iron phosphate peak, and the main diffraction peaks are basically consistent with the X-ray diffraction peaks of lithium iron phosphate studied by Yangluo Gui et al. [23], Chaironi Latif et al. [24], Lingyu Guan et al. [25] and Chaoqi shen et al. [26]. The space group is Pnma and it belongs to an orthorhombic olivine structure. In addition, there are also magnesium oxide diffraction peaks. The main diffraction peaks are (111) [27], (200) [28], (220), etc., which all correspond to the standard card (PDF#75-1525). The crystallinity is good, indicating that the magnesium oxide and lithium iron phosphate carbon composites have been successfully synthesized. The peaks are similar to the X-ray diffraction patterns of LiFePO_4_ and MgO alone in the lithium iron phosphate electrode material prepared by Xiankun Huang et al. [29] by introducing magnesium oxide and other substances as additives, with sharp X-ray diffraction peaks. The XRD average lattice constant of the composite (Table 1) shows that the average lattice constant of each composite is basically the same as that of the standard pure phase lithium iron phosphate (PDF#83-2092), indicating that the introduction of magnesium oxide has no significant effect on the crystal structure of lithium iron phosphate.

Moreover, the small figure on the right in Figure 3 shows the offset diagram of the main diffraction peak of lithium iron phosphate (311). It can be found that the offset of the main diffraction peak of composite samples with different magnesium oxide content has little change, indicating that the introduction of magnesium oxide has no great effect on the crystal structure of lithium iron phosphate. In addition, for X-ray diffraction analysis, the Bragg equation [30] is:2d · sinθ = *n* · λ(1)

The d in the equation is the crystal plane spacing of the crystal. θ is the diffraction angle of X-ray. λ is the wavelength of X-ray. n is an integer. Due to diffraction angle θ being basically unchanged, and n and λ are for a specific value, according to the Bragg equation [30], the crystal plane spacing d value of its crystal is also basically unchanged. For the diffraction pattern, the uneven bottom line may be caused by amorphous, low crystalline and amorphous carbon [31,32]. It can be seen from Figure 4 that the average lattice parameters b-axis and c-axis of the composite basically decrease with the increase in magnesium oxide content.

The XRD diffraction pattern of zinc oxide and lithium iron phosphate carbon composites is showed in Figure 5. It shows that there are lithium iron phosphate peaks in each composite, and the main diffraction peaks, including (111), (211), and (311), correspond to the standard card (PDF#83-2092), indicating that there is a complete pure phase lithium iron phosphate peak for each composite sample, which is basically consistent with the lithium iron phosphate X-ray diffraction peaks studied by Ming Shi et al. [33] and A.Sarmadi et al. [34]. The space group is Pnma. It belongs to the orthorhombic olivine structure.

In addition, there are also zinc oxide diffraction peaks. The main diffraction peaks are (111), (200) [35,36], (220), (311), and (222), which correspond to the standard card (PDF #77-0191) one by one, and the crystallinity is good, indicating that the composite of ZnO and LiFePO_4_/C composites has been successfully synthesized. This is similar to the X-ray diffraction pattern of the zinc oxide-modified lithium iron phosphate prepared by Shuxin Liu et al. [8], which has the complete diffraction peak of lithium iron phosphate and zinc oxide. In addition, with the increase of the mass ratio of ZnO to LiFePO_4_/C, the diffraction peak of lithium zinc phosphate appears when the mass ratio reaches 0.16. For the diffraction pattern, the uneven bottom line may be caused by amorphous, low crystalline and amorphous carbon [31,32]. The XRD average lattice constant of the composite (Table 2) shows that the average lattice constant of each composite is basically unchanged, compared with that of the standard pure phase lithium iron phosphate (PDF#83-2092), indicating that the introduction of zinc oxide does not greatly change the crystal structure of lithium iron phosphate [37]. It can be seen from Figure 6 that the average lattice parameters b-axis and c-axis of the composite have a similar change trend. There is basically no change before the mass ratio is 0.15. When the mass ratio continues to increase, they slightly decrease and then increase.

### 2.3. Infrared Spectrum Analysis

The Fourier infrared spectrum analysis of MgO and LiFePO_4_/C composite with different mass ratios is shown in Figure 7. It shows that the samples of magnesium oxide and lithium iron phosphate carbon composites with six different mass ratios mainly show obvious infrared absorption characteristic peaks around 3432, 1631, 1384, 1138, 1061, 971, 639, 578, 550, 507 and 475 cm^−1^ [38,39]. It can also be seen from Figure 7 that with the increase of magnesium oxide content, the vibration absorption peak with a wave number of about 971 cm^−1^ tends to shift slightly to the left. According to the research of P. Jozwiak and Roger Frech [40,41], the infrared vibration absorption peak of lithium iron phosphate in the wave number range of about 400~1200 cm^−1^ is mainly caused by the vibration of the P-O bond and Li-O bond. The group vibration modes of materials include symmetrical stretching vibration (v_1_) [42], antisymmetric stretching vibration (v_3_) [43], symmetrical bending vibration (v_2_) and antisymmetric bending vibration (v_4_) [44]. The P-O bond in lithium iron phosphate mainly corresponds to symmetric stretching vibration (v_1_) and antisymmetric stretching vibration (v_3_), while the O-P-O bond mainly corresponds to symmetric bending vibration (v_2_) and antisymmetric bending vibration (v_4_) [44]. It can be seen that the infrared analysis images of the six composite samples show absorption peaks at wave numbers of about 1138 cm^−1^ and 1061 cm^−1^, which are mainly caused by the antisymmetric stretching vibration (v_3_) of the P-O bond, and the absorption peak at the wave number of about 971 cm^−1^ is mainly caused by the symmetrical stretching vibration (v_1_) of the P-O bond. The absorption peaks with wave numbers of about 639, 578, 550, 507 and 475 cm^−1^ are mainly caused by the mixed vibration of symmetric bending vibration (v_2_) and antisymmetric bending vibration (v_4_) of the O-P-O bond [45].

To sum up, the wave number is about 372~1139 cm^−1^, which mainly corresponds to the internal vibration mode of the PO_4_^3−^ ion [46]. For the infrared spectrum absorption peak caused by the vibration of the lithium ion (Li^+^), it mainly appears in the low wavenumber section with a wavenumber of about 400~600 cm^−1^ [47], but because some infrared absorption peaks caused by the vibration of PO_4_^3-^ also exist in the low wavenumber section with the wavenumber of about 400~600 cm^−1^, the vibration absorption peak in this wavenumber section is the superposition of the two [45]. Therefore, it is impossible to clearly distinguish which peaks are infrared absorption peaks caused by lithium ion (Li^+^) vibration. The infrared vibration absorption peak of the Mg-O bond appears at about 870, 598, 440 cm^−1^ [48,49], indicating that there is an infrared characteristic peak of magnesium oxide. In general, the vibration absorption peak patterns of the infrared spectra of six MgO and LiFePO_4_/C composite samples with different mass ratios are basically consistent with the FTIR test peak patterns of lithium iron phosphate by Heru Setyawan et al. [50], with vibration absorption peaks corresponding to lithium iron phosphate.

The Fourier infrared spectra of the ZnO and LiFePO_4_/C composite are shown in Figure 8. It shows that the samples of zinc oxide and lithium iron phosphate carbon composites with six different mass ratios mainly show obvious infrared absorption characteristic peaks around 1061, 971, 639, 578, 550, 507 and 475 cm^−1^ [51,52]. It is mainly the vibration absorption peak of the P-O bond and the O-P-O bond [45]. The infrared vibration absorption peak of the Zn-O bond appears at about 500 cm^−1^ [53,54] and 600 cm^−1^ [55], indicating that there is an infrared characteristic peak of zinc oxide. In general, the vibration absorption peak patterns of the infrared spectra of six zinc oxide and lithium iron phosphate carbon composite samples with different mass ratios are basically consistent with the FTIR test peak patterns of lithium iron phosphate by Heru Setyawan et al. [50], with vibration absorption peaks corresponding to lithium iron phosphate.

### 2.4. Scanning Electron Microscope Analysis

Scanning electron microscope (SEM) analysis is a very important analysis method for the measurement and analysis of the micro surface morphology, particle size and particle dispersion of materials [56]. In order to analyze the micro morphology of magnesium oxide and lithium iron phosphate carbon composites with different mass ratios, the samples with a mass ratio of MgO to LiFePO_4_/C of 0.09 were analyzed by SEM. The results are shown in Figure 9, which shows that the particles of the composites are dispersed relatively evenly. Through the statistics of particle size, it is found that the particle size distribution is similar to the normal distribution. The particle size is mainly concentrated at about 300 nm. The particle size is smaller than that of some samples (400–800 nm) in reference [57], and it is smaller than the particle size of some samples (micron) in reference [58]. In addition, it can be seen from Figure 9 that there are many smaller particles around the larger particles, and the smaller particles may be magnesium oxide, indicating that the introduction of magnesium oxide may hinder the growth of composite particles to a certain extent, and the smaller material particles may have higher conductivity.

The SEM analysis result with the mass ratio of ZnO to LiFePO_4_/C of 0.16 is shown in Figure 10. It can be seen that the particles of the composite are relatively evenly dispersed. Through the statistics of the particle size, it is found that the particle size distribution follows a quasi-normal distribution, and the particle size is mainly concentrated at about 400 nm. Compared with zinc oxide-modified lithium iron phosphate particles (about 1 µm) prepared by Shuxin Liu et al. [8] and some lithium iron phosphate sample particles (2–50 μm) prepared by Marco G. Rigamonti et al. [59], our particle size is smaller. It shows that the introduction of zinc oxide may hinder the growth of composite particles to a certain extent and improve the electromagnetic properties of composites.

### 2.5. Magnetic Performance Analysis

The magnetic properties of MgO and LiFePO_4_/C composites were analyzed by a vibration magnetometer (VSM). The results are shown in Figure 11 and Figure 12 and Table 3. Table 3 shows that the magnetization of all composites is greater than the room temperature magnetization (about 0.35 emu/g) calculated by theoretical simulation [60], indicating that there is a gap between theory and practice.

When magnesium oxide is not introduced, the saturation magnetization of lithium iron phosphate carbon composites is smaller than that of reference [61] (the maximum is about 1 emu/g), and the magnetization of composites with magnesium oxide is larger than that of reference [61], indicating that the introduction of magnesium oxide enhances the magnetic properties of composites. The magnetic properties of lithium iron phosphate carbon composites with different contents of magnesium oxide have great changes. It can be seen from the variation trend of magnetic parameters (Figure 12) that the saturation magnetization (Ms, emu/g), residual magnetization (Mr, emu/g) and hysteresis loop area of the hysteresis loop (kOe·emu/g) basically increase with the increase in magnesium oxide, while the coercivity Hc first increases and then decreases. For lithium iron phosphate, its Neel temperature T_N_ is 50 K [62,63]. That is, it is antiferromagnetic when the temperature is lower than 50 K, and paramagnetic when the temperature is higher than 50 K. The six samples analyzed at room temperature should be paramagnetic, the hysteresis loop of theory should be a straight line, and the hysteresis loop area should be zero. It can be seen from the results that the hysteresis loop area is small, while the coercivity of the sample is relatively large, showing weak ferromagnetism, which may be caused by ferrous or ferromagnetic impurities after the introduction of magnesium oxide [13,14,15,16,17,18,19,20,21,22,23,24,25,26,27,28,29,30,31,32,33,34,35,36,37,38,39,40,41,42,43,44,45,46,47,48,49,50,51,52,53,54,55,56,57,58,59,60,61,62,63,64]. However, no obvious ferromagnetic impurity phase was found in the previous XRD analysis, indicating that the ferromagnetic impurity content may be small, and its XRD diffraction peak may be relatively weak. It is difficult to find its obvious XRD characteristic peak, which is similar to the analysis of some samples of lithium iron phosphate carbon composites synthesized by Ming Chen et al. [13], and there is no obvious impurity peak in XRD. Weak ferromagnetism is shown in the hysteresis loop of the sample at room temperature.

Figure 13 is the hysteresis loops of ZnO and LiFePO_4_/C composites at room temperature, and Table 4 and Figure 14 show the variation of magnetic parameters. It can be found that the magnetic properties of lithium iron phosphate carbon composites with different contents of zinc oxide have great changes. It can be seen from the variation trend chart of magnetic parameters (Figure 14) that the Ms and Mr basically increase with the increase of magnesium oxide, the area of hysteresis loop basically does not change, and the Hc first increases slightly and then decreases. Compared with other references [60,61], the lithium iron phosphate carbon composite with zinc oxide has relatively large saturation magnetization, indicating that the introduction of zinc oxide is conducive to improving the magnetic properties of the composite.

## 3. Experiment

### 3.1. Experimental Steps

The experimental steps include two parts. The first part is the synthesis of LiFePO_4_/C samples. The samples of LiFePO_4_/C composites are prepared by sol-gel. Firstly, a carbon source (ethylene glycol), a lithium source (lithium hydroxide monohydrate), a phosphorus source (ammonium dihydrogen phosphate) and an iron source (ferric nitrate nine hydrate) are added with an appropriate amount of distilled water according to the material ratio of 1:1:1:1, mixed and dissolved in the beaker, sealed with fresh-keeping film and aged at room temperature for 20 h, and then the mixed solution is stirred in a 60 °C water bath until dry, placed in an 80 °C drying oven for full drying, and then ground with a mortar. Finally, LiFePO_4_/C composites were obtained by calcining in a muffle furnace at 850 °C for 15 h under the protection of activated carbon powder.

The second part is synthesizing AO-type metal oxides (MgO, ZnO) and LiFePO_4_/C composites with different mass ratios. The detailed steps are as follows:

(1) Weigh the mass of corresponding drugs (lithium iron phosphate, magnesium nitrate, zinc nitrate and ethylene glycol) according to the set proportion, in which the set mass ratio x of magnesium oxide and lithium iron phosphate is 0.00, 0.03, 0.06, 0.09, 0.12 and 0.15, respectively; the mass ratio of ethylene glycol to magnesium nitrate is 0.2482, and the mass ratio x of zinc oxide to lithium iron phosphate is 0.00, 0.05, 0.10, 0.16, 0.21 and 0.27, respectively; and the mass ratio of ethylene glycol to zinc nitrate is 0.4341. Then, dissolve the ethylene glycol solution in an appropriate amount of distilled water to obtain solution A, metal nitrate (magnesium nitrate or zinc nitrate) in an appropriate amount of distilled water to obtain solution B, lithium iron phosphate in an appropriate amount of distilled water to obtain solution C, and then drop ethylene glycol solution A into metal nitrate (magnesium nitrate or zinc nitrate) solution B and continuously stir with a glass rod to obtain the mixed solution D.

(2) Drop solution C into mixture D and stir it continuously with a glass rod to obtain mixture E.

(3) Use the agitator to stir mixture E in a 40 °C water bath until the wet gel is obtained, stirring for about 5 h.

(4) Use a large beaker to cover the sample beaker, then put it into a drying oven and dry gel at 40℃. The average drying time is 10 h.

(5) Grind the sample with a mortar to obtain the powder.

(6) Put the powder into a three-layer ceramic crucible protected by carbon.

(7) Calcine the sample in a three-layer ceramic crucible protected by carbon in a muffle furnace.

(8) After calcination, grind the sample with a mortar again to obtain the final sample powder.

### 3.2. Characterization of Material Properties

For the phase structure analysis of AO-type metal oxides (MgO, ZnO) and LiFePO_4_/C composites with different contents, the crystal structure of the materials was characterized by X-ray diffractometer (D/max-2500v/pc, Rigaku, Tokyo, Japan), and Cu-Kα was used as target radiation. The scanning speed was set to 10°/min, and the scanning angle range was set to 2θ = 10~80°. For the Fourier infrared spectrum analysis of AO-type metal oxides (MgO, ZnO) and LiFePO_4_/C composites with different contents, the Spectrum Two (PerkinElmer) infrared spectrum analyzer (FT-IR) (PerkinElmer, Waltham, MA, USA) was used to test and analyze the possible functional groups and internal chemical bonds in the samples prepared by the experiment. The area of the infrared spectrum test was set as 400~4000 cm^−1^. For the microstructure, particle size, degree of uniform dispersion and agglomeration of AO-type metal oxide (MgO, ZnO) and LiFePO_4_/C composites with different contents, ZEISS EVO18 (ZEISS, Jena, Germany) was used to analysis the micro morphology, and a Nano Measurer 1.2 w (Fudan University, Shanghai, China) used to statistically analyze the particle size of the material. The magnetic properties of AO-type metal oxides (MgO, ZnO) and LiFePO_4_/C composites with different contents were tested by a VSM-100 vibrating sample magnetometer (YINGPU MAGNETOELECTRIC company, Changchun, China). During the analysis, the maximum magnetic field was H = 8000 Oe (0.8 T), the field increasing step moment was 4, and the maximum sensitivity of the instrument was 5 × 10^−5^ emu. The hysteresis loop of the sample was analyzed at room temperature, and various magnetic performance parameters of the sample, including area of hysteresis loop, residual magnetization (Mr), coercivity (Hc) and saturation magnetization (Ms), could be obtained.

## 4. Conclusions

In this paper, AO-type metal oxides (MgO and ZnO) and LiFePO_4_/C composites were successfully prepared by a two-step method. The effect of AO-type metal oxides (MgO and ZnO) on LiFePO_4_/C composites was studied. MgO and LiFePO_4_/C composites were prepared by a two-step sol-gel method. The thermal properties, phase structure, functional groups, chemical bonds, micro surface morphology and magnetic properties of the samples during the synthesis of MgO and LiFePO_4_/C composites with different mass ratios were studied by TG, XRD, FTIR, SEM and VSM. The research shows that there are three obvious mass change platforms for the decrease of the heating mass of the composite precursor. After 400 °C, the mass of the precursor remains basically unchanged, and composite materials may begin to be synthesized. The final composite material is about 70.5% of the total mass of the precursor. The main peaks of lithium iron phosphate and magnesium oxide appeared in the phase structure analysis of the MgO and LiFePO_4_/C composites, and there was no obvious impurity peak. Infrared spectrum analysis shows that the composites have infrared vibration absorption peaks of lithium iron phosphate and magnesium oxide. SEM analysis of micro morphology shows that the composite has obvious particles and good dispersion, and the particle size is mainly 300 nm. The hysteresis loop analysis at room temperature shows that the saturation magnetization (Ms, emu/g), residual magnetization (Mr, emu/g) and area of hysteresis loop of the composites increase with the increase of magnesium oxide, while the coercivity (Hc, Oe) first increases slightly and then decreases, and the maximum Ms is 11.11 emu/g. ZnO and LiFePO_4_/C composites were synthesized by a two-step sol-gel method and analyzed by TG, XRD, FTIR, SEM and VSM. The TG analysis shows that there are three obvious mass change platforms for the decrease of the heating mass of the composite precursor. After 410 ℃, the mass of the precursor remains basically unchanged, and the composite may begin to be synthesized. The final composite is about 69% of the total mass of the precursor. The phase structure analysis of ZnO and LiFePO_4_/C composites showed that the main peaks of lithium iron phosphate and zinc oxide are found in the XRD figure. Infrared spectrum analysis shows that infrared vibration absorption peaks of lithium iron phosphate and zinc oxide are found in the FTIR figure of composites. SEM analysis shows that there are obvious particles and good dispersion, and the particle size is mainly 400 nm. The hysteresis loop measurement at room temperature shows that the introduction of zinc oxide makes the Ms and Mr of the composite basically increase with the increase of zinc oxide, while the area of the hysteresis loop remains stable after a slight increase. The Hc first increases slightly and then decreases with the increase of zinc oxide.

## Figures and Tables

**Figure 1 molecules-28-01970-f001:**
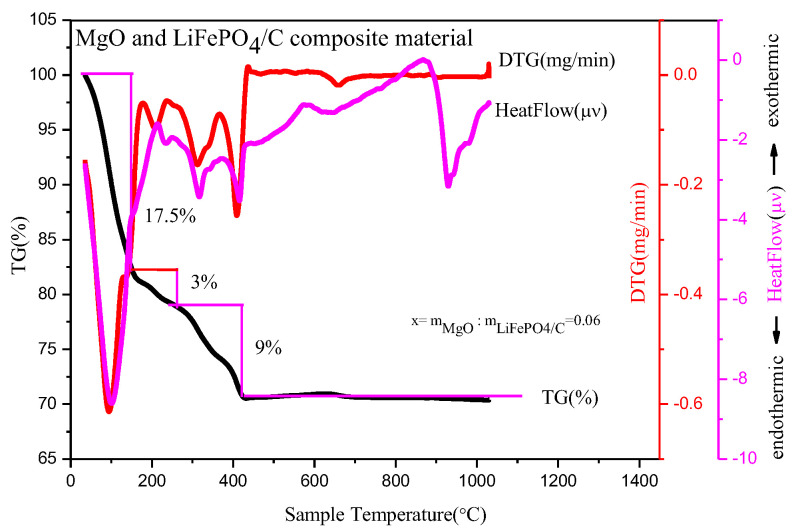
Thermogravimetric analysis of composite material with mass ratio of MgO to LiFePO_4_/C equal to 0.06.

**Figure 2 molecules-28-01970-f002:**
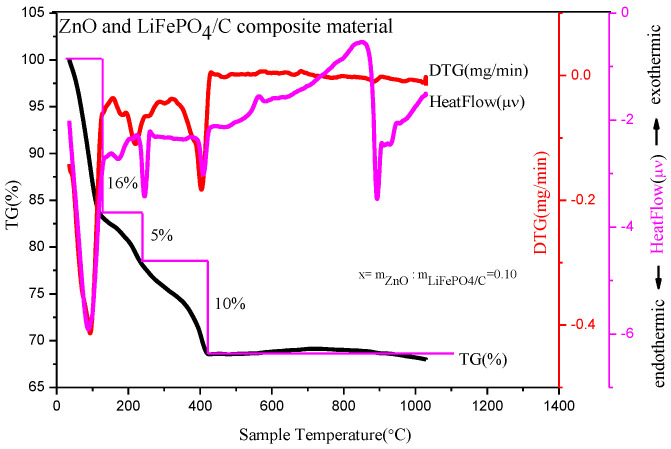
Thermogravimetric analysis of composites with mass ratio of ZnO to LiFePO_4_/C composite equal to 0.10.

**Figure 3 molecules-28-01970-f003:**
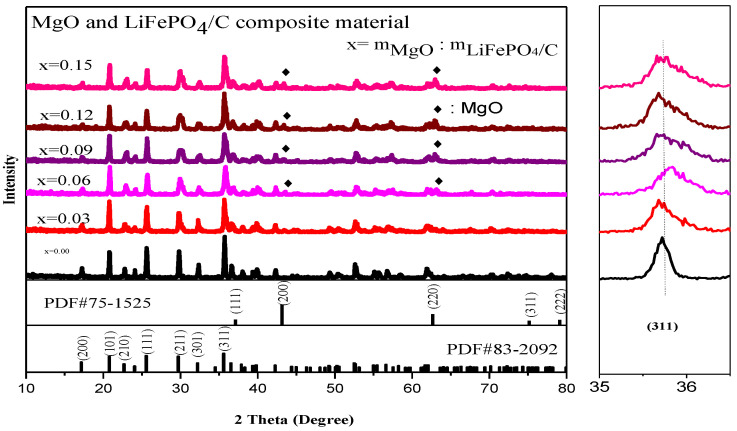
X-ray diffraction analysis of MgO and LiFePO_4_/C composites.

**Figure 4 molecules-28-01970-f004:**
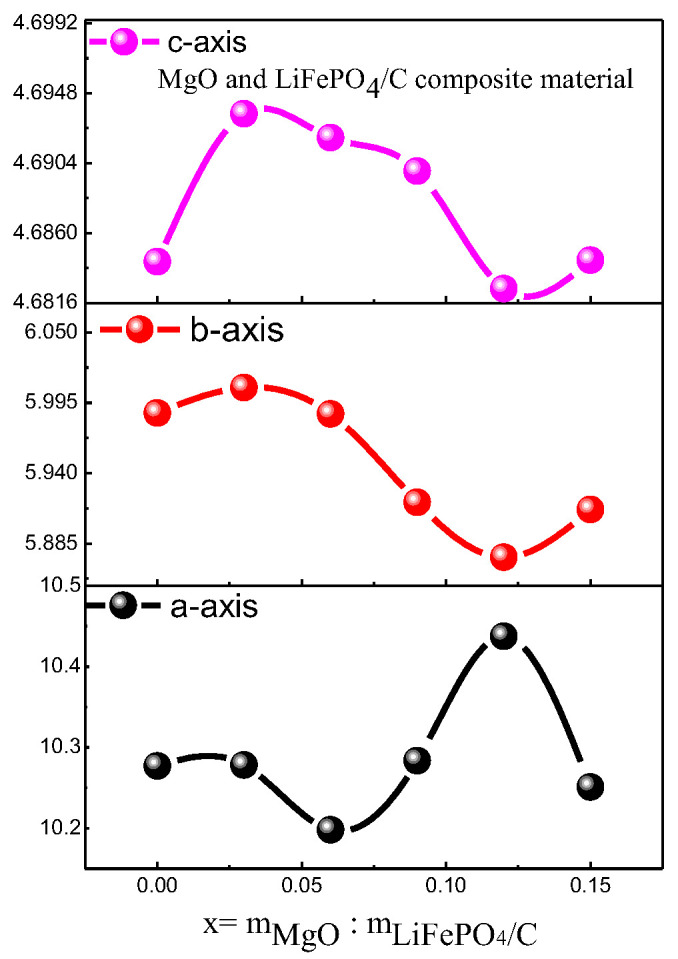
Variation trend of XRD parameters of MgO and LiFePO_4_/C composites.

**Figure 5 molecules-28-01970-f005:**
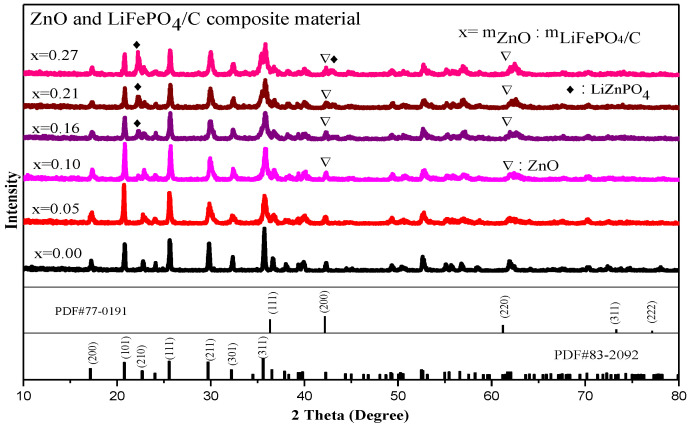
XRD analysis of ZnO and LiFePO_4_/C composites.

**Figure 6 molecules-28-01970-f006:**
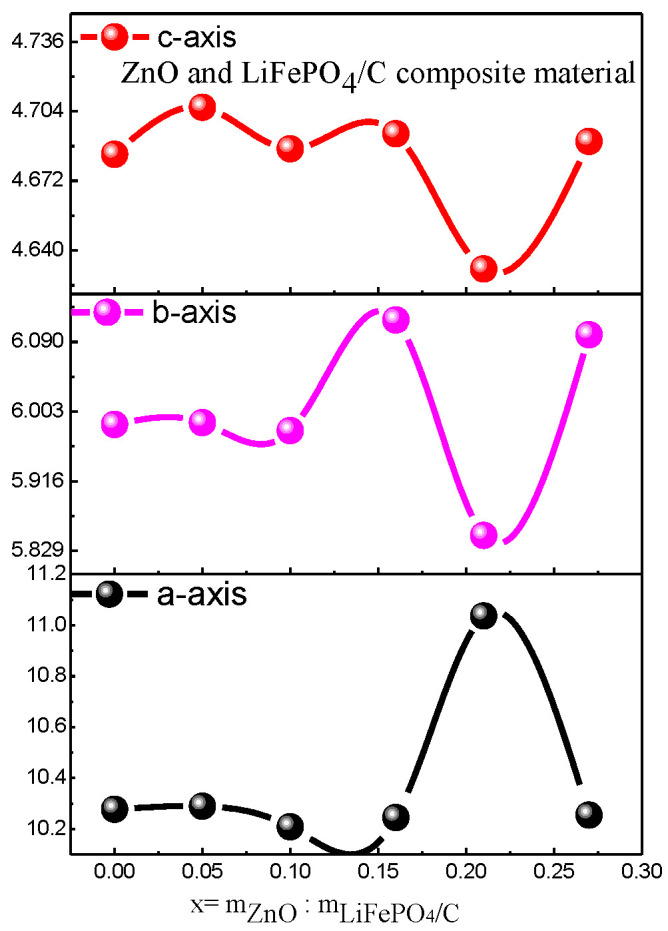
Variation trend of XRD parameters of ZnO and LiFePO_4_/C composites.

**Figure 7 molecules-28-01970-f007:**
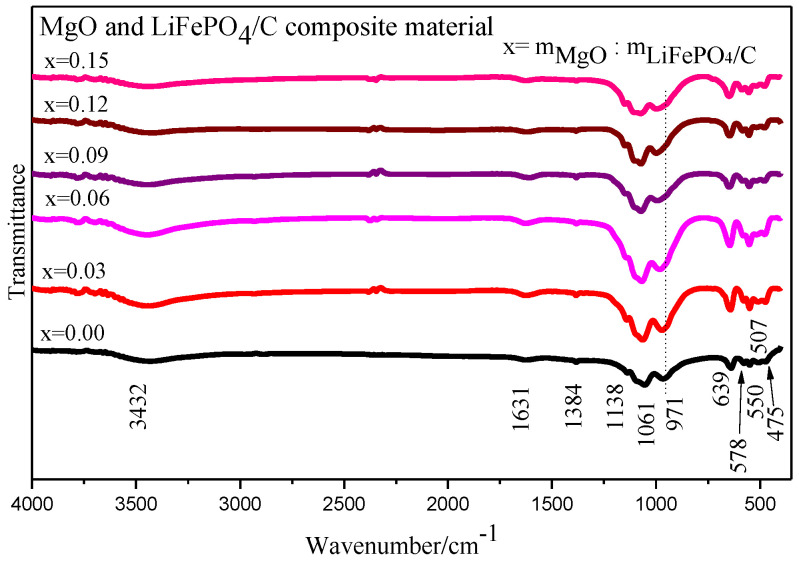
Infrared spectrum analysis of MgO and LiFePO_4_/C composites.

**Figure 8 molecules-28-01970-f008:**
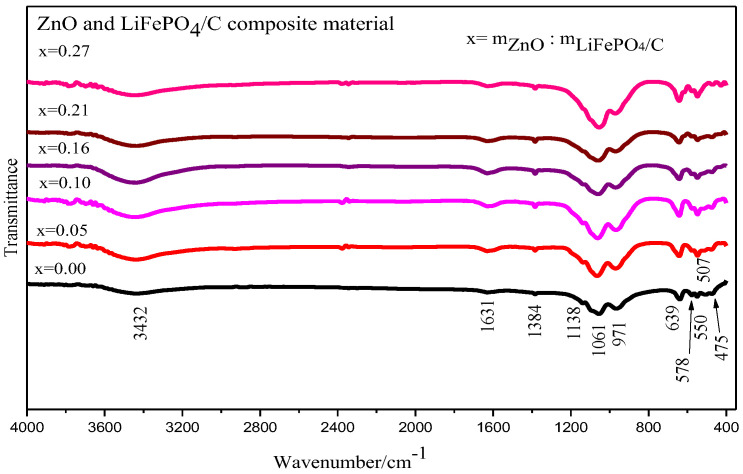
Infrared spectrum analysis of ZnO and LiFePO_4_/C composites.

**Figure 9 molecules-28-01970-f009:**
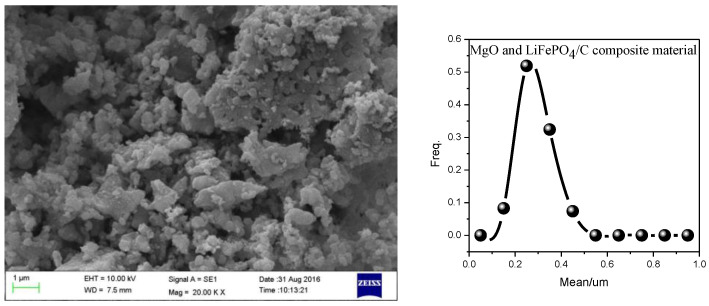
SEM analysis of MgO and LiFePO_4_/C composites (the mass ratio of MgO to LiFePO_4_/C is 0.09. The calcination temperature is 850 °C and the calcination time is 15 h.).

**Figure 10 molecules-28-01970-f010:**
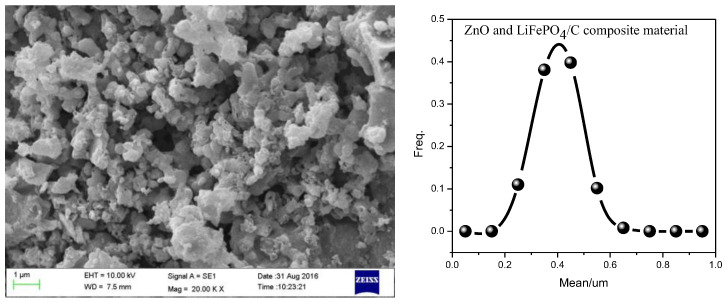
SEM analysis of ZnO and LiFePO_4_/C composites (the mass ratio of ZnO to LiFePO_4_/C is 0.16. The calcination temperature is 850 °C and the calcination time is 15 h).

**Figure 11 molecules-28-01970-f011:**
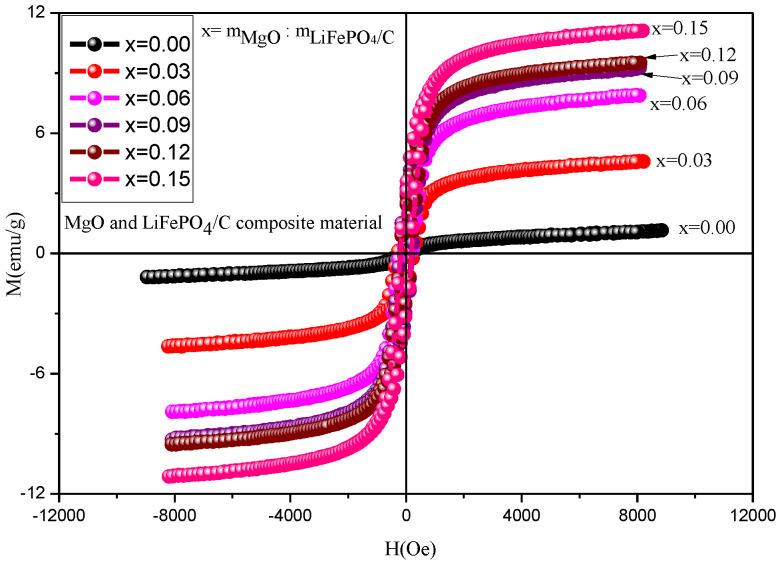
Hysteresis loop of MgO and LiFePO_4_/C composites at room temperature.

**Figure 12 molecules-28-01970-f012:**
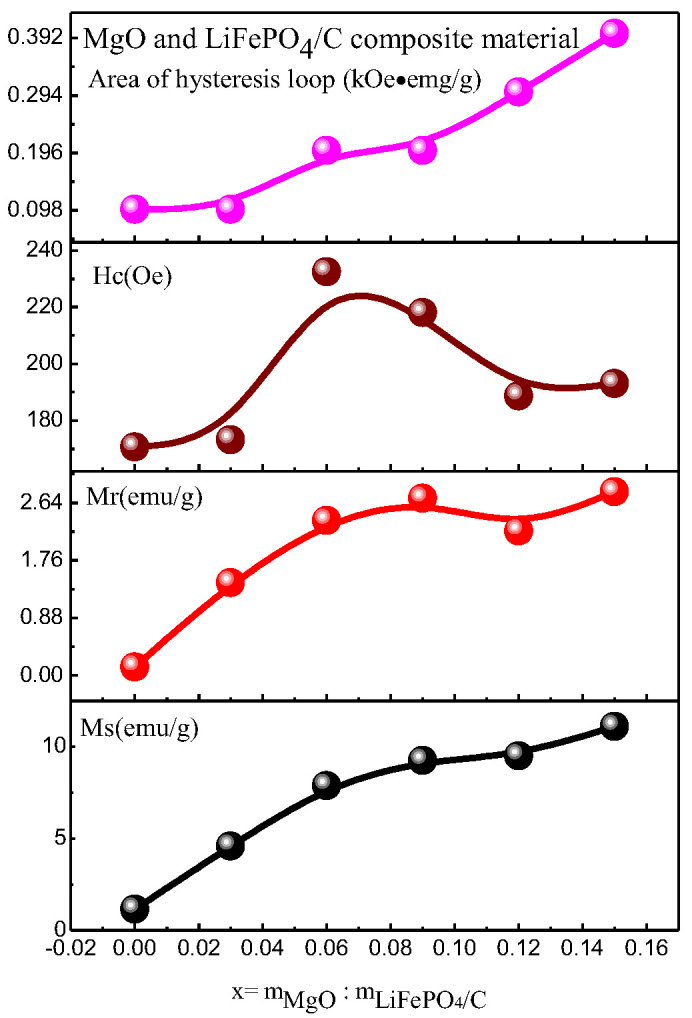
Variation trend of magnetic parameters of MgO and LiFePO_4_/C composites at room temperature.

**Figure 13 molecules-28-01970-f013:**
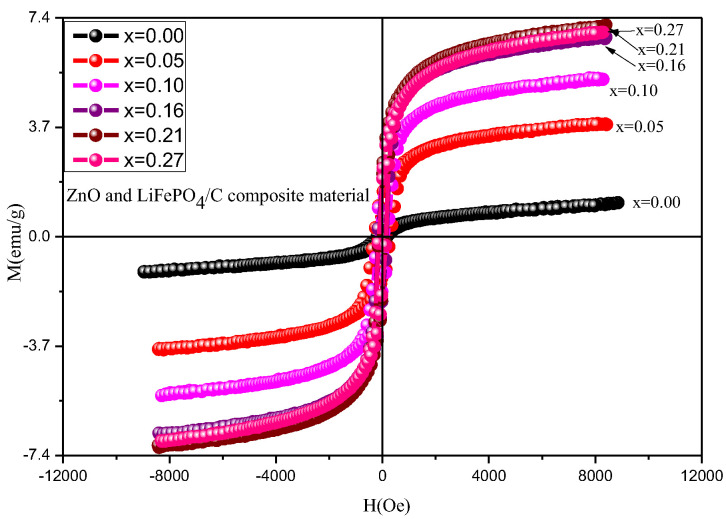
Hysteresis loop of ZnO and LiFePO_4_/C composites at room temperature.

**Figure 14 molecules-28-01970-f014:**
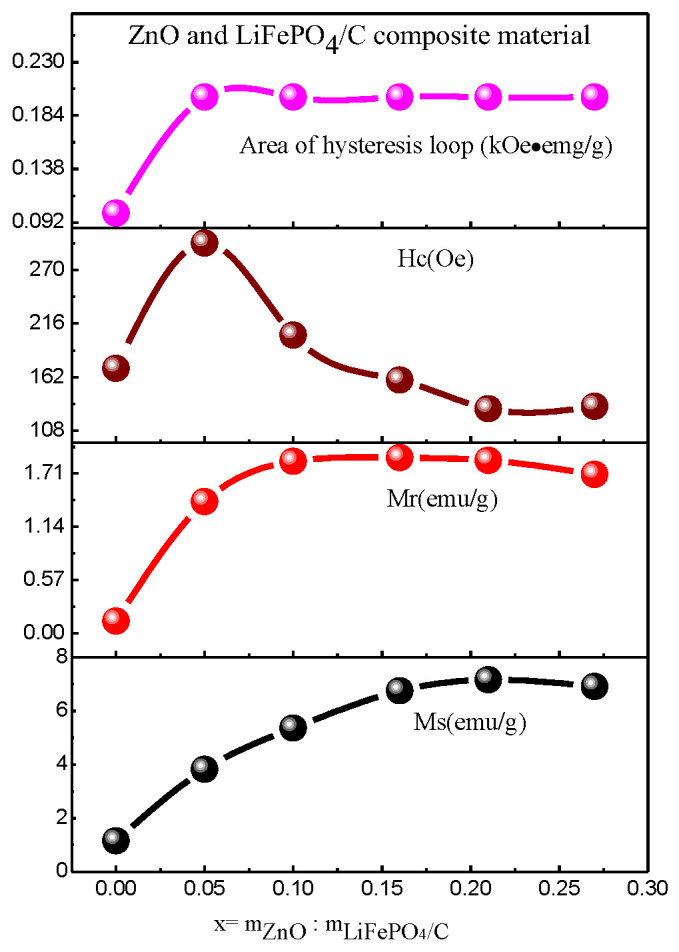
Variation trend of magnetic parameters of ZnO and LiFePO_4_/C composites at room temperature.

**Table 1 molecules-28-01970-t001:** XRD analysis parameters of magnesium MgO and LiFePO_4_/C composites.

Samples	$x=m_{MgO}:m_{LiFePO_{4}}$	Average Lattice Constants		
		a-Axis (Å)	b-Axis (Å)	c-Axis (Å)
PDF#75-1525		4.1900	4.1900	4.1900
PDF#83-2092		10.3340	6.0100	4.6930
13L2F2P15h851	0.00	10.2769	5.9870	4.6842
cMg4h651	0.03	10.2780	6.0069	4.6935
cMg4h652	0.06	10.1981	5.9862	4.6920
cMg4h653	0.09	10.2834	5.9173	4.6899
cMg4h654	0.12	10.4374	5.8742	4.6825
cMg4h655	0.15	10.2511	5.9115	4.6843

**Table 2 molecules-28-01970-t002:** XRD analysis parameters of ZnO and LiFePO_4_/C composites.

Samples	$x=m_{ZnO}:m_{LiFePO_{4}}$	Average Lattice Constants		
		a-Axis (Å)	b-Axis (Å)	c-Axis (Å)
PDF#77-0191		4.2800	4.2800	4.2800
PDF#83-2092		10.3340	6.0100	4.6930
13L2F2P15h851	0.00	10.2769	5.9870	4.6842
cZn4h651	0.05	10.2892	5.9891	4.7059
cZn4h652	0.10	10.2081	5.9791	4.6867
cZn4h653	0.16	10.2453	6.1175	4.6937
cZn4h654	0.21	11.0362	5.8485	4.6315
cZn4h655	0.27	10.2542	6.0989	4.6902

**Table 3 molecules-28-01970-t003:** Magnetic parameters of MgO and LiFePO_4_/C composites at room temperature.

Samples	x	Ms (emu/g)	Mr (emu/g)	Hc (Oe)	Area Of Hysteresis Loop (kOe·emu/g)
13L2F2P15h851	0.00	1.15	0.13	170.67	0.1
cMg4h651	0.03	4.59	1.42	173.22	0.1
cMg4h652	0.06	7.89	2.37	232.55	0.2
cMg4h653	0.09	9.26	2.71	218.07	0.2
cMg4h654	0.12	9.50	2.21	188.63	0.3
cMg4h655	0.15	11.11	2.81	193.00	0.4

**Table 4 molecules-28-01970-t004:** Magnetic parameters of ZnO and LiFePO_4_/C composites at room temperature.

Samples	x	Ms (emu/g)	Mr (emu/g)	Hc (Oe)	Area of Hysteresis Loop (kOe·emu/g)
13L2F2P15h851	0.00	1.15	0.13	170.67	0.1
cZn4h651	0.05	3.82	1.41	296.87	0.2
cZn4h652	0.10	5.36	1.84	204.37	0.2
cZn4h653	0.16	6.75	1.88	159.21	0.2
cZn4h654	0.21	7.16	1.85	130.25	0.2
cZn4h655	0.27	6.91	1.70	132.72	0.2

## Data Availability

Not applicable.

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
