# Peer review of "Structure and Magnetic Properties of AO and LiFePO4/C Composites by Sol-Gel Combustion Method"

_molecules, 2023, doi:10.3390/molecules28041970_

Round 1
Reviewer 1 Report
The manuscript “Structure and magnetic properties of AO and LiFePO4/C composites by sol-gel combution method” reports about the synthesis and characterization of composite materials containing LiFePO4 and MgO and ZnO. The materials were characterized by spectroscopic and morphological techniques such as IR, SEM, XRD, TGA and DSC.
The language of manuscript needs to be carefully revised since many typing and grammar errors are spread in the main text (for example: “The experimental steps is made up of two parts”)
What does “combution” mean? Is it for “combustion”?
Why did the author report that “different carbon sources” were used, but only ethylene glycol is cited?
FTIR was used as a vibrational spectroscopy, I suggest to the author to couple this technique with a more efficient (for this kind of materials) analysis method, that is Raman spectroscopy.
Author Response
The manuscript “Structure and magnetic properties of AO and LiFePO4/C composites by sol-gel combution method” reports about the synthesis and characterization of composite materials containing LiFePO4 and MgO and ZnO. The materials were characterized by spectroscopic and morphological techniques such as IR, SEM, XRD, TGA and DSC.
The language of manuscript needs to be carefully revised since many typing and grammar errors are spread in the main text (for example: “The experimental steps is made up of two parts”)
1.What does “combution” mean? Is it for “combustion”?
Authors’ response:
These have been corrected accordingly.
It is “sol-gel combustion method”.
2.Why did the author report that “different carbon sources” were used, but only ethylene glycol is cited?
Authors’ response:
These have been corrected accordingly.
It is modified as “carbon source (ethylene glycol)”.
FTIR was used as a vibrational spectroscopy, I suggest to the author to couple this technique with a more efficient (for this kind of materials) analysis method, that is Raman spectroscopy.
Authors’ response:
These have been corrected accordingly.
FTIR is a very effective analytical method for chemical bonds.
Reviewer 2 Report
The paper is good for the publishing in Molecules journal.
There are some minor revisions:
1) Lines 48 and 378 indicate the Nair temperature. Neel?
2) In Figures 9 and 10, you should make a more detailed caption of the figure, indicate what kind of diagram, please.
3) In figures 11 and 13, it is necessary to indicate the measurement temperature, on the graph or in the caption to the figure.
4) In line 392, the numbering of figures is violated, instead of Fig 5.12, you should write Fig.12
5) In line 393, replace MgO with ZnO, please.
Author Response
The paper is good for the publishing in Molecules journal.
Thank you for your kind comments. We revised the manuscript in accordance with the reviewers’ comments, and carefully proof-read the manuscript to minimize typographical, and grammatical errors.
Here below is our description on revision according to the comments.
1) Lines 48 and 378 indicate the Nair temperature. Neel?
Authors’ response:
These have been corrected accordingly.
It is “Neel temperature”.
2) In Figures 9 and 10, you should make a more detailed caption of the figure, indicate what kind of diagram, please.
Authors’ response:
These have been corrected accordingly.
It is modified as:
Fig. 9 SEM analysis of MgO and LiFePO4/C composites ( the mass ratio of MgO to LiFePO4/C is 0.09. The calcination temperature is 850℃ and the calcination time is 15 hours.)
Fig. 10 SEM analysis of ZnO and LiFePO4/C composites (the mass ratio of ZnO to LiFePO4/C is 0.16. The calcination temperature is 850℃ and the calcination time is 15 hours.)
3) In figures 11 and 13, it is necessary to indicate the measurement temperature, on the graph or in the caption to the figure.
Authors’ response:
These have been corrected accordingly.
It is modified as:
Fig. 11 Hysteresis loop of MgO and LiFePO4/C composites at room temperature
Fig. 12 Variation trend of magnetic parameters of MgO and LiFePO4/C composites at room temperature
Fig. 13 Hysteresis loop of ZnO and LiFePO4/C composites at room temperature
4) In line 392, the numbering of figures is violated, instead of Fig 5.12, you should write Fig.12
Authors’ response:
These have been corrected accordingly.
It is modified as:
Fig. 12 Variation trend of magnetic parameters of MgO and LiFePO4/C composites at room temperature
5) In line 393, replace MgO with ZnO, please.
Authors’ response:
These have been corrected accordingly.
It is modified as:
Fig. 13 is the hysteresis loops of ZnO and LiFePO4/C composites at room temperature, and Tab. 4 and Fig. 14 show the variation of magnetic parameters.